# Extracellular Vesicles Derived from Mesenchymal Stromal Cells Delivered during Hypothermic Oxygenated Machine Perfusion Repair Ischemic/Reperfusion Damage of Kidneys from Extended Criteria Donors

**DOI:** 10.3390/biology11030350

**Published:** 2022-02-22

**Authors:** Teresa Rampino, Marilena Gregorini, Giuliana Germinario, Eleonora Francesca Pattonieri, Fulvia Erasmi, Maria Antonietta Grignano, Stefano Bruno, Esra Alomari, Stefano Bettati, Annalia Asti, Marina Ramus, Mara De Amici, Giorgia Testa, Stefania Bruno, Gabriele Ceccarelli, Nicoletta Serpieri, Carmelo Libetta, Vincenzo Sepe, Flavia Blasevich, Federica Odaldi, Lorenzo Maroni, Francesco Vasuri, Gaetano La Manna, Matteo Ravaioli

**Affiliations:** 1Department of Nephrology, Dialysis and Transplantation, Fondazione IRCCS Policlinico San Matteo, University of Pavia, 27100 Pavia, Italy; t.rampino@smatteo.pv.it (T.R.); ef.pattonieri@smatteo.pv.it (E.F.P.); fulvia.erasmi89@gmail.com (F.E.); wta87@hotmail.it (M.A.G.); annalia.asti@unipv.it (A.A.); marina.ramus01@universitadipavia.it (M.R.); n.serpieri@smatteo.pv.it (N.S.); carmelo.libetta@unipv.it (C.L.); v.sepe@smatteo.pv.it (V.S.); 2Department of Internal Medicine and Therapeutics, University of Pavia, 27100 Pavia, Italy; 3Department of General Surgery and Transplantation, IRCCS, Azienda Ospedaliero-Universitaria di Bologna, 40138 Bologna, Italy; giuliana.germinario2@unibo.it (G.G.); fede.odaldi@gmail.com (F.O.); lorenzo.maroni@aosp.bo.it (L.M.); matteo.ravaioli6@unibo.it (M.R.); 4Dipartimento di Scienze Mediche e Chirurgiche (DIMEC), University of Bologna, 40126 Bologna, Italy; 5Department of Food and Drug, University of Parma, 43124 Parma, Italy; stefano.bruno@unipr.it (S.B.); esraa.alomari@iu.edu.jo (E.A.); 6Biopharmatec TEC, University of Parma, Tecnopolo Padiglione 33, 43124 Parma, Italy; stefano.bettati@unipr.it; 7Department of Medicine and Surgery, University of Parma, 43125 Parma, Italy; 8Laboratory of Immuno-Allergology of Clinical Chemistry and Pediatric Clinic, Fondazione IRCCS Policlinico S. Matteo, 27100 Pavia, Italy; M.DeAmici@smatteo.pv.it; 9Department of Pediatrics, Fondazione IRCCS Policlinico San Matteo, University of Pavia, 27100 Pavia, Italy; g.testa@smatteo.pv.it; 10Department of Medical Sciences and Molecular Biotechnology Center, University of Torino, 10126 Torino, Italy; stefania.bruno@unito.it; 11Human Anatomy Unit, Department of Public Health, Experimental and Forensic Medicine, University of Pavia, 27100 Pavia, Italy; gabriele.ceccarelli@unipv.it; 12Department of Neuroimmunology and Neuromuscular Diseases, Fondazione IRCCS Neurological Institute Carlo Besta, 20133 Milan, Italy; flavia.blasevich@istituto-besta.it; 13“F. Addarii” Institute of Oncology and Transplantation Pathology, S. Orsola-Malpighi University Hospital, 40138 Bologna, Italy; francesco.vasuri@aosp.bo.it; 14Department of Nephrology, S.Orsola-Malpighi Hospital, University of Bologna, 40138 Bologna, Italy; gaetano.lamanna@unibo.it

**Keywords:** ischemia/reperfusion injury, kidney transplantation, machine perfusion, expanded criteria donors, mesenchymal stromal cells, extracellular vesicles, COXIV-1, caspase 3, HGF, VEGF

## Abstract

**Simple Summary:**

In this study, we explore for the first time an innovative tool for organ preservation aimed to preventing ischemia reperfusion injury (IRI) in marginal kidneys from expanded criteria donors (ECD) unsuitable for transplantation. Ex vivo hypothermic oxygenated perfusion (HOPE) with and without MSC-derived EV and normothermic reperfusion (NR) with artificial blood composed of bovine hemoglobin were applied on kidneys to evaluate global renal ischemic damage score, renal ultrastructure, mitochondrial distress, apoptosis, cell proliferation index, and the mediators of energy metabolism. Our study demonstrates that kidney conditioning with HOPE+EV arrests the ischemic damage, prevents reoxygenation-dependent injury, and preserves tissue integrity. EV delivery during HOPE can be considered a new organ preservation strategy to increase the donor pool and improving transplant outcome. The originality of our study lies an EV and HOPE combined novel setting use in kidneys from ECD, but also in any condition for graft dysfunction such as ischemia/reperfusion.

**Abstract:**

The poor availability of kidney for transplantation has led to a search for new strategies to increase the donor pool. The main option is the use of organs from extended criteria donors. We evaluated the effects of hypothermic oxygenated perfusion (HOPE) with and without extracellular vesicles (EV) derived from mesenchymal stromal cells on ischemic/reperfusion injury of marginal kidneys unsuitable for transplantation. For normothermic reperfusion (NR), we used artificial blood as a substitute for red blood cells. We evaluated the global renal ischemic dam-age score (GRS), analyzed the renal ultrastructure (RU), cytochrome c oxidase (COX) IV-1 (a mitochondrial distress marker), and caspase-3 renal expression, the tubular cell proliferation index, hepatocyte growth factor (HGF) and vascular endothelial growth factor (VEGF) tissue levels, and effluent lactate and glucose levels. HOPE+EV kidneys had lower GRS and better RU, higher COX IV-1 expression and HGF and VEGF levels and lower caspase-3 expression than HOPE kidneys. During NR, HOPE+EV renal effluent had lower lactate release and higher glucose levels than HOPE renal effluent, suggesting that the gluconeogenesis system in HOPE+EV group was pre-served. In conclusion, EV delivery during HOPE can be considered a new organ preservation strategy for increasing the donor pool and improving transplant outcome.

## 1. Introduction

Kidney transplantation represents the gold-standard treatment for patients with end-stage renal disease. Unfortunately, the huge gap between the number of transplant candidates and organ availability has had a significant impact on patient morbidity and mortality. Although new therapies such as mesenchymal stromal cells (MSC) and drug monitoring have improved graft outcomes [1,2,3], new strategies are needed for increasing the donor pool. At present, the main option is to use marginal kidney allografts from older donors with comorbidities such as hypertension, mild renal impairment, and death from cerebrovascular events, defined as extended criteria donors (ECD) [4,5].

However, ECD organs are more exposed to ischemic/reperfusion injury (IRI) compared to those from standard donors. Consequently, their use presents an increased risk of primary non-function, delayed graft function, and decreased graft survival [6,7,8,9,10].

IRI pathophysiology consists of two phases: the initial ischemic phase involving interrupted blood flow to the organs, and the reperfusion phase, in which the blood flow is restored. During ischemia, there is a switch from aerobic to anaerobic metabolism, which leads to intracellular changes: ATP levels decline, intracellular protons and calcium increase; there is cytoskeletal, mitochondrial, and endoplasmic reticulum dysfunction and cellular swelling occurs. Although restoring the blood flow is essential for reversing the damage caused by ischemia, it may also result in reoxygenation-dependent injury due to reactive oxygen species (ROS) increase, mitochondrial and cellular membrane peroxidation, the recruitment of immune and inflammatory cells, and activation of the programmed cell death pathways. Combined with low energy storage, all of these events can lead to irreversible injury of marginal kidneys, unless rapidly treated [11,12,13,14,15,16].

To increase the number of successful transplants, the use of machine perfusion (MP) is currently proposed for assessing organ viability prior to transplantation, but also to attenuate organ damage, limiting the discard rate [17,18,19,20,21]. In fact, it has been proven in several animal models that MP has a protective role against apoptosis by reducing caspase activation and for improving circulation [22]. The main effects of MP are the removal of waste products and inflammatory mediators and the supplementation of metabolic substrates for generating ATP and glutathione, which protect against ROS [23,24,25,26].

At present, dynamic MP can be performed in hypothermic (HMP) or normothermic (NMP) conditions with or without oxygen. Both perfusion systems increase the survival of organs harvested from ECD [21,27,28,29,30,31].

Recently, we have shown that hypothermic oxygenated machine perfusion (HOPE) for ECD grafts is safe and protects organs from IRI by restoring ATP levels, improving function recovery and kidney and liver transplantation outcomes [32,33,34,35,36].

The addition of hemoglobin-based oxygen carriers (HBOCs) [37] in the perfusion solution under NMP conditions increases tissue oxygenation. Several NMP experiments have shown that HBOCs yield similar, if not better, results in comparison to red blood cells (RBCs) in the liver [38] and kidney [39].

Dynamic MP also presents the possibility of delivering targeted therapies to organs before transplantation. Using a kidney donation after circulatory death (DCD) rat model, we have demonstrated that MSC/MSC-derived extracellular vesicles (EV) delivered during HMP protected kidneys from ischemic injury. In particular, we observed the up regulation of genes encoding enzymes that improve cell energy metabolism and ion membrane transport in MSC EV-treated kidneys [40]. Several other studies have shown the beneficial effects of MSC/MSC-derived EV in IRI models. They accelerate renal recovery after damage, promoting cell proliferation and blocking inflammation and apoptosis. [3,41,42,43,44,45,46] Based on these data, we hypothesized that the combined use of HOPE and EV, i.e., the delivery of EV during HOPE, may enhance their individual beneficial effects on IRI. In the present study, we explored for the first time an innovative tool for organ preservation aimed at preventing IRI damage in vulnerable marginal kidneys.

We applied ex vivo HOPE to which MSC-derived EV had been added as a preservation technique in discarded organs from ECD after brain death. We also designed a normothermic reperfusion (NR) phase using HBOCs as RBC substitutes to explore whether EV action may also prevent reoxygenation-dependent injury.

## 2. Materials and Methods

### 2.1. Study Design

This single-center, prospective, interventional arm pilot randomized clinical study applied MSC-derived EV therapy combined with MP in discarded human kidneys from ECD. This study was performed at the General Surgery and Transplant Unit of the IRCCS, Azienda Ospedaliero-Universitaria of Bologna in accordance with the Helsinki Declaration. All experimental procedures were approved by the Ethics in Research Committee in June 2016. The study has been registered at ClinicalTrials.gov (ClinicalTrials.gov ID: NCT03837197) (Figure 1).

### 2.2. Eligibility

Kidneys deemed ineligible for transplant were recruited consecutively in the study from June 2019 to June 2020. The inclusion criteria were: marginal kidney pairs considered non-eligible for transplantation, with histological Remuzzi score ≥5 [47]. The exclusion criteria were: kidneys eligible for single or double kidney transplantation, kidneys not technically compatible with reperfusion due to anatomical vascular and/or excretory anomalies, kidneys from donors with serological positivity for anti- hepatitis C virus (HCV) antibodies and/or seropositive for Human immunodeficiency virus (HIV), and all cases of kidneys with non-calculable infectious donation risk.

### 2.3. Organ Retrieval

Organs were procured using the technique developed by Starzl [48]. Following aortic clamping, the abdominal organs were flushed in situ through the aorta with cold Celsior or Belzer solution (Bridge to Life, Columbia, SC, USA), retrieved, dipped in a bag filled with preservation fluid, and stored on ice. Pre-transplant biopsies were performed ac-cording to our retrieval protocol. The retrieved organs were stored on ice during the trans-fer from the donor to the transplant hospital, during the biopsy analysis, until cross-matched results were returned, and until the final decision of non-eligible kidneys.

### 2.4. Randomization

One kidney from the pair was randomized in a 1:1 ratio and preserved with HOPE + MSC-derived EV (HOPE+EV study group) or with HOPE without MSC-derived EV (HOPE control group) after static cold storage (SCS) during transportation to the hospital. In both groups, the preservation was followed by perfusion with HBOCs. Randomization was carried out through Medidata Balance (New York, NY, USA) and was performed after the organ had been deemed unsuitable for transplantation.

### 2.5. Ex Vivo Hypothermic Oxygenated Perfusion

HOPE was performed using a VitaSmart^®^ device designed for abdominal organ perfusion (Bridge to Life). The kidneys were connected to the perfusion device through sterile disposable tubes and the vessels were cannulated with a cannula specified for vascular size. HOPE was performed for 4 h using Belzer MPS^®^ UW Machine Perfusion Solution (Bridge to Life) at 4 °C through the artery with peristaltic flow at 25–30 mmHg pressure. Flow, pressure, and temperature were monitored and registered automatically by the minute. Oxygen and carbon dioxide partial pressure (pO2 and pCO2) and pH was monitored every 15 min in the effluent perfusate via blood gas analysis (GEM Premier 4000, Werfen, Italy).

### 2.6. Ex Vivo Kidney Reperfusion with HBOCs

After ex vivo kidney perfusion, the kidneys of both groups were reperfused in a model simulating transplant reperfusion. The organs were kept on ice until they were placed again on the reperfusion device. After that, they were perfused using HBOCs.

The circuit was primed with one unit of HBOCs; the organs were reperfused for 4 h at 20–25 °C and 60 mmHg pressure through peristaltic flow with the same MP procedure. As in the perfusion phase, during ex vivo reperfusion, we monitored the flow, pressure, and temperature, which were registered automatically by the minute and every 15 min in the effluent perfusate via blood gas analysis (GEM Premier 4000).

The effluent perfusate was collected before and after the HMP and NMP to quantify the glucose and lactate concentrations via blood gas analysis (GEM Premier 4000).

Biopsy samples were collected after SCS and after both hypothermic and normothermic preservation. The kidney tissue was fixed in 10% formalin for morphological and immunohistochemical studies and in 2.5% glutaraldehyde and cacodylate sodium buffer (pH 7.4) for ultrastructural analysis.

### 2.7. MSC and EV Isolation and Characterization

Bone marrow MSC were purchased from Lonza (Basel, Switzerland) and cultured in a MSC basal medium BulletKit (Lonza). The cells were used within the sixth passage and the typical mesenchymal markers were detected by cytofluorimetric analysis as de-scribed previously [49,50,51,52].

EV were obtained by ultracentrifugation [50,51]. Briefly, the EV were obtained from 24 mL of the supernatants of MSC cultured overnight in Roswell Park Memorial Institute (RPMI) 1640 medium. The cell debris and apoptotic bodies were removed by 20-min centrifugation at 3000× *g* and by microfiltration using a 0.22-mm vacuum filter unit (Millipore, Billerica, MA, USA). EV were purified by 2-h ultracentrifugation at 100,000× *g* at 4 °C (Beckman Coulter Optima L-100 K, Fullerton, CA, USA). EV were used fresh or stored at −80 °C after resuspension in RPMI 1640 medium supplemented with 1% dimethyl sulfoxide (Sigma, St. Louis, MO, USA).

EV were characterized using a human cytofluorimetric bead-based MACSPlex exosome kit (Miltenyi Biotec, Bergisch Gladbach, Germany) according to the manufacturer’s proto-col [53]. EV were diluted in MACSPlex buffer (MPB) in the presence of the MACSPlex exosome capture beads. EV on the capture beads were counterstained by adding APC-conjugated anti-CD9, anti-CD63, and anti-CD81 detection antibodies and were incubated for 1 h at room temperature in the dark on an orbital shaker at 450 rpm. Post-incubation, the beads were washed first with 1 mL MPB at 3000× *g* for 5 min, followed by a longer washing step by incubating the beads in 1 mL MPB on an orbital shaker (as before) for 15 min. Then, the beads were centrifuged at 3000× *g* for 5 min and the supernatant was carefully aspirated, leaving a residual volume of 150 mL per tube for acquisition. Flow cytometry was performed using a CytoFLEX flow cytometer (Beckman Coulter, Brea, CA, USA), where approximately 5000 single-bead events were recorded per sample. The median fluorescence intensity (MFI) for all 39 exosomal markers was corrected for background and gated based on their respective fluorescence intensity as per the manufacturer’s instructions.

The concentration of the EV preparations was determined using a NanoSight LM10 instrument (NanoSight Ltd., Amesbury, UK) equipped with a 405 nm laser. The recordings of three 60-s videos were examined using nanoparticle tracking analysis (NTA version 3.4).

### 2.8. Hemoglobin-Based Oxygen Carriers

Glutathione-polymerized bovine hemoglobin (bHb) was prepared using a modified version of published protocols [54]. Fresh bovine blood was collected, using EDTA as an anticoagulant (1.5 mg/mL blood). The RBCs were recovered by 10-min centrifugation at 15,000× *g* and washed three times by centrifugation/resuspension cycles with sterile isotonic saline solution (0.9% NaCl). The cells were lysed with a PandaPlus 200 homogenizer (GEA^®^, Parma, Italy) for one cycle at 500 torr. Cell debris were removed by 1-h centrifugation at 10,000× *g* at 4 °C. Oxygenated bHb was reacted with glutaraldehyde in a 1:10 molar ratio at 37 °C for 3 h under continuous stirring. The reaction was quenched at 4 °C for 30 min with a two-fold molar excess of NaBH_4_ dissolved in 20 mM phosphate-buffered saline (PBS), pH 8. Diafiltration through a sterile tangential flow filter was performed at 4 °C for 24 h to remove unreacted glutaraldehyde and to exchange the buffer to sterile Ringer’s solution (7.2 g/L NaCl, 0.017 g/L CaCl_2_, KCl 0.37 g/L, pH 7.4). The HBOC solution at a final concentration of 3.5 g/dL was frozen and stored at −80 °C. The oxygen binding parameters of the final preparation were measured using a tailor-made tonometer [55,56] at 25 °C, the intended temperature of use. The polymerized bHb exhibited a P50 (Hb oxygen affinity) of 8.2 torr and lost its binding cooperativity (Hill coefficient = 1.0) as reported previously [57]. Oxidation to methemoglobin (met-Hb) was followed spectroscopically at every preparation step and after the perfusion experiments. No significant met-Hb formation was observed.

### 2.9. Renal Morphology

Five non-consecutive cross-sections of each kidney were stained with periodic acid–Schiff (PAS) and examined by two investigators in double-blind fashion using a Nikon Eclipse E200 microscope (Amsterdam, the Netherlands) connected to a CCD (charge-coupled de-vice) camera and Image J imaging analysis software (NIH, Bethesda, MD, USA). Renal damage was assessed by scoring all tubules acquired in each high-powered field (HPF). We analyzed at least 10 non-consecutive fields. The tubular lesions analyzed were tubular epithelial cell flattening (TF), brush border loss (BBL), bleb formation (BF), tubular necrosis (TN), and tubular lumen obstruction (TO). TF and BBL were classified as mild lesions; BF, TN, and TO were severe lesions.

The global ischemic renal damage score as described by Paller et al. [14] was obtained by assigning each lesion a score: TF (1 point), BBL (1 point), BF (2 points), TN (2 points), and TO (2 points). When ≥2 lesions were present in the same tubule, the most severe score was assigned.

### 2.10. Transmission Electron Microscopy

Transmission electron microscopy (TEM) was performed using a standard technique: kidney tissues from all groups were immediately fixed in 2.5% glutaraldehyde and cacodylate sodium buffer (pH 7.4) for 2 h at room temperature. Then, the samples were rinsed in cacodylate sodium buffer (pH 7.4) overnight and post-fixed in 1% aqueous osmium (OsO4) for 90 min at room temperature. The samples were dehydrated in increasing ethanol concentrations (50% to 100%). After polymerization with epoxy resin (Epon 812), the blocks were sectioned on a microtome (Leica, Wetzlar, Germany) to generate semithin sections. The sections were counterstained with toluidine blue for optical microscopy. Thin sections were double-stained with uranyl acetate and lead citrate. Observations and micrographs were performed on an electron microscope (Jeol JEM 1200, Tokyo, Japan) operating at 100 kV.

### 2.11. Tubular Cell Proliferation Index and COX IV-1 Renal Expression

The tubular cell proliferation index (IPT) was defined as the ratio between the nuclei ex-pressing proliferating cell nuclear antigen (PCNA) and the total nuclei in each tubule. The paraffin-embedded tissue sections were collected on poly-L-lysine-coated slides (Dako, Carpinteria, CA, USA), dewaxed in xylol, cleared in a decreasing series of alcohol, and re-hydrated with distilled water. Endogenous peroxidase was blocked with H2O2 (3.7% vol/vol for PCNA and 3.0% vol/vol for cytochrome c oxidase [COX] IV-1) followed by H2O for 15 min. After three washes in 150 mM PBS, the sections underwent microwave antigen retrieval. Subsequently, they were exposed overnight at 4 °C to monoclonal mouse anti-PCNA antibody (1:200, Santa Cruz Biotechnology, Santa Cruz, CA, USA) or to COX IV-1 anti-human antibody (1:50, ABclonal, Woburn, MA, USA). After three washes in PBS, the immunocomplex was visualized with biotin-streptavidin-peroxidase complex and 3,3-diaminobenzidine (Dako, Glostrup, Denmark). The sections were lightly counter-stained with Harris hematoxylin. Negative controls were established by omitting the primary antibody and substituting immunoglobulin G (IgG) for the primary antibodies.

Ten non-consecutive sections from each immunostained kidney were analyzed. The images were captured using a Nikon Eclipse E200 microscope connected to a CCD camera and ImageJ (NIH).

The IPT was evaluated by counting the number of PCNA-positive nuclei/total nuclei of each tubule in every field analyzed (×40).

COX IV-1 expression was assessed by converting the immunohistochemistry image to black and white and quantifying the pixel number (×10). The results are shown as the percentage of black pixel numbers/total pixel numbers [58].

### 2.12. HGF and VEGF Tissue Levels

Human vascular endothelial growth factor (VEGF) and human hepatocyte growth factor (HGF) were evaluated using a commercial enzyme-linked immunosorbent assay kit (R&D Systems, Minneapolis, MN, USA) following the manufacturer’s instructions. The results are expressed as pg/mL.

### 2.13. Apoptosis

The dewaxed and rehydrated tissue sections were permeabilized by 5-min incubation with PBS–0.1% Triton X-100 at room temperature.

After washing in PBS, the sections were incubated with blocking buffer (PBS–0.1% Tween 20 + 5% goat serum) in a humidified chamber for 2 h at room temperature. Then, the sections were washed again in PBS and incubated with primary antibody (1:200 dilution in blocking buffer, rabbit anti-caspase-3 antibody, active, C8487, Sigma) overnight in a humidified chamber at 4 °C. The following day, the slides were incubated with 3.7% H2O2 for 15 min in a humidified chamber at room temperature to inhibit endogenous peroxidase. The secondary antibody was horseradish peroxidase-labeled polymer conjugated to goat anti-rabbit antibody (EnVision + System (DAB), DAKO North America, Carpinteria, CA, USA). The immunocomplex was visualized with 3,3-diaminobenzidine (Dako, Glostrup, Denmark). The sections were lightly counterstained with Harris hematoxylin. Negative controls were established by omitting the primary antibody and substituting IgG for the primary antibodies.

Ten non-consecutive sections of each immunostained kidney were analyzed. The images were captured using a Nikon Eclipse E200 microscope connected to a CCD camera and ImageJ (NIH).

Apoptosis was evaluated by counting the number of caspase-3-positive cells in every HPF analyzed (×40).

### 2.14. Statistical Analysis

GraphPad Prism software (San Diego, CA, USA) was used for the statistical analyses. Data are expressed as the mean and standard deviation or the median and interquartile range (IQR) or as the minimum (min) and maximus (max) in accordance with the data distribution. Parametric and non-parametric variables among the two groups were com-pared using Student’s unpaired t-test or the Mann-Whitney test. Continuous and non-continuous variables among >2 groups were compared using analysis of variance (ANOVA) or Kruskal-Wallis test as appropriate followed by the Tukey test and Dunn’s multiple comparisons test, respectively. A *p*-value < 0.05 was considered statistically significant.

## 3. Results

### 3.1. Donor Characteristics

Table 1 shows the donor characteristics: age, sex, weight, height, body mass index (BMI), blood group, cause of death, type of donation, histological Remuzzi score of kidneys en-rolled in the study, and duration of SCS before hypothermic perfusion. Ten kidneys from five ECD were included in the study. Five kidneys were perfused with HOPE (HOPE) and the other ones with EV(HOPE+EV). Subsequently all kidneys were reperfused with HBOCs (NR HOPE *n* = 5; NR HOPE +EV *n* = 5).

### 3.2. EV Characterization

Cytofluorimetric analysis revealed that the EV were positive for the typical MSC markers such as CD29, CD44, CD49e, CD105, CD146, and for the exosomal tetraspanins CD9, CD63, and CD81, but did not express hematopoietic (CD45, CD3, CD8), endothelial (CD31), or epithelial (CD24 and CD326) markers (Appendix A) as previously reported (53). NTA revealed that the EV size was in the typical range of 30–300 nm (Appendix A). We used 28.5 × 10^9^ EVs to treat the perfused kidney

### 3.3. Perfusion Parameters

The perfusion parameters did not differ between the groups during HMP and NR (Table 2). The perfusion and reperfusion phases were performed without complications. The solutions containing the EV and HBCOs did not affect the perfusion parameters.

### 3.4. Global Renal Ischemic Damage Score

The global ischemic renal damage score was significantly more severe in SCS-preserved ECD kidneys (median, 70; range, 41–99) than in HOPE- (median, 40; range, 20–60) or HOPE+EV-preserved kidneys (median, 24; range, 21–36) (*p* < 0.001). Tubular epithelial cell flattening, blebs, tubular necrosis, and lumen obstruction were more frequent in HOPE than in HOPE+EV kidneys (Figure 2).

### 3.5. Ultrastructural Analysis

Figure 3 shows the representative TEM renal sections of all groups. The SCS group had loss of normal cell architecture, cell shrinkage and necrosis, and dysmorphic and swollen mitochondria with loss and distortion of cristae. The HOPE group had partial loss of cell architecture and the mitochondria appeared spheroidal and elongated. In contrast, the HOPE+EV group had significantly reduced damage: the cell architecture and mitochondrial morphology were well preserved and the nucleus and basal membrane were normal. After NR, the HOPE group had cytoplasmic vacuolization and brush borders with short, spaced microvilli, while the HOPE+EV group had cells with normal morphology and organelles, and interestingly, a long, preserved brush border.

### 3.6. COX IV-1 Renal Expression

To evaluate mitochondrial distress, we analyzed COX IV-1 renal expression. COX IV-1 renal expression was higher in HOPE+EV kidneys than in HOPE kidneys. COX IV-1 renal expression was highest in the NR HOPE+EV group (*p* < 0.0001) (Figure 4).

### 3.7. Caspase-3 Renal Expression

To evaluate apoptosis, we examined caspase-3, a crucial mediator of programmed cell death. Caspase-3 renal expression was higher in the HOPE groups than in the HOPE+EV groups (*p* < 0.001) (Figure 5A).

### 3.8. Tubular Cell Proliferation Index

To evaluate tissue viability, we studied tubular cell proliferation. The IPT after NMP and HMP was higher in the HOPE+EV groups than in the HOPE groups (*p* < 0.0001) (Figure 5B).

### 3.9. HGF and VEGF Tissue Levels

The NR HOPE+EV group had increased renal tissue levels of the growth factors HGF and VEGF (HGF: median, 5.85 pg/mL; range, 2.56–9.15; VEGF: median, 1.95 pg/mL; range, 0.83–3.13) compared to the NR HOPE group (HGF: median, 0.86 pg/mL; range, 0.33–1.4; VEGF: median, 0.30 pg/mL; range, 0.05–0.78, *p* < 0.001).

### 3.10. Lactate and Glucose Levels in Effluent Fluid

To evaluate tissue performance, we measured the main intermediators of energy metabolism in the effluent: lactate and glucose.

At the end of the HOPE the lactate levels significantly increased (*p* < 0.05) and glucose levels decreased in the effluent of both groups, suggesting the conversion of glucose, provided by the Belzer solution, to lactate as in anaerobic glycolysis (Table 3).

During NR, we observed lower lactate release in the HOPE+EV effluent than in the HOPE effluent. In parallel, the HOPE+EV group had increased effluent glucose levels than the HOPE group (expressed as the Tend/T0 ratio). As glucose was not provided by the HBOC solution during the normothermic reperfusion, the higher glucose levels indicate that it was synthesized by the renal tissue, suggesting that the gluconeogenesis system was well preserved (Figure 6).

## 4. Discussion

Organ conditioning before transplantation is a technique with great potential by virtue of its ability to transfer drugs and buffer to the tissue. Here, we report the results of an innovative and pioneering EV delivery system for human kidney preservation in organs discarded due to high chronic damage scores. EV have recently gained interest for drug delivery purposes since they offer many advantages, such as high permeability, low immunogenicity, and non-cytotoxicity [59,60,61]. Moreover, among the different regenerative medicine strategies, EV have been recognized as a promising and innovative tool for accelerating tissue recovery after organ damage. They improve graft survival due to their ability to transfer to damaged cells proteins, bioactive lipids, microRNAs, mRNAs, and growth factors, which exert proliferative, anti-fibrotic, and immunomodulatory effects [62,63,64,65,66,67].

Here, we used an original ex vivo IRI model where ECD organs were perfused with MSC-derived EV delivered during HOPE, and re-perfused with a solution containing HBOCs.

EV delivery during HOPE significantly improved organ quality even though the kidneys were organs discarded for transplantation due to their high chronic histological score.

The switch from aerobic to anaerobic metabolism induced by prolonged ischemia leads to intracellular changes and architecture subversion. We observed the typical morphological changes such as brush border loss, bleb formation, tubular epithelial cell flattening, tubular necrosis, and tubular lumen obstruction in all groups. The HOPE+EV kidneys had significantly lower global renal damage score and better-preserved tissue ultra-structure than the HOPE and SCS kidneys. To evaluate the effect on oxygen-induced renal damage, we used a solution containing HBOCs lacking the immune/inflammatory cellular components of blood.

HBOCs are based on cell-free Hb subjected to either chemical or genetic modifications to increase retention in the bloodstream while maintaining oxygen delivery to tissues [35,56,68,69]. They include conjugated, crosslinked, polymerized, encapsulated Hb of either human or animal origin [70,71,72,73,74,75,76,77,78,79,80,81]. Depending on the preparation protocol, they may exhibit various biochemical and biophysical properties [82] that can be finely regulated to achieve the optimal product for specific applications. Unlike RBCs, HBOCs can be used at different temperatures, their properties can be finely regulated, they do not need blood typing, and they can be stored for much longer times either as a solution or frozen, i.e., all important advantages for the experimenter. Using a rat model, we have previously demonstrated that EV delivered during organ hypothermic perfusion protected the kidney from ischemic injury [38], but we did not study the NR phase.

Under ischemia/reperfusion, hypoxia and oxidative stress lead to the overproduction of mitochondrial ROS, Ca2+ overload, and prolonged mitochondrial permeability transition pore (mPTP) opening that forms a channel to release cytochrome c into the cytoplasm and then activate the apoptosis cascade [83,84]. COX is a multi-subunit enzyme complex that couples the transfer of electrons from cytochrome c to molecular oxygen and contributes to a proton electrochemical gradient across the inner mitochondrial membrane. The complex consists of 13 mitochondrial- and nuclear-encoded subunits. The COX IV-1 sub-unit is a component of COX, the last enzyme in the mitochondrial electron transport chain, which drives oxidative phosphorylation. COX IV-1 degradation is induced in cells exposed to hypoxia [85]. Here, NR HOPE kidneys had lower renal COX IV-1 expression than NR HOPE+EV kidneys at the end of NR, suggesting that HOPE+EV preservation is more protective against IRI damage to the mitochondrial structure. This mitochondrial dysfunction is also accompanied by structural remodeling and morphological shape changes (mitochondrial spheroids). Using TEM, we confirmed that the HOPE and SCS kidneys had subverted cell architecture and more mitochondrial spheroids compared with the HOPE+EV and NR HOPE+EV kidneys (Figure 3). The mitochondria can mediate cell death through the release of proapoptotic proteins. In particular, once cytochrome c is translocated to the cytosol, it binds to the APAF-1 protein, which then triggers the activation of the caspase cascade [86]. Accordingly, we investigated caspase expression in renal tissue and observed significantly fewer apoptotic nuclei in HOPE+EV kidneys than in HOPE kidneys, confirming the known protective effect of EV against apoptosis [67,87].

The greater viability of the EV-conditioned kidneys was also demonstrated by the higher IPT in the HOPE+EV kidneys and even more in the NR HOPE+EV kidneys than in HOPE kidneys. In parallel, VEGF and HGF tissue levels were higher in the NR HOPE+EV group than in the NR HOPE group, confirming that the EV regenerative effect may be mediated by the release of HGF and VEGF [41,45,67,88].

Finally, the analysis of the principal intermediaries in energetic metabolism in the effluents of the perfused kidneys confirmed the beneficial effects of EV on organ viability. During hypothermic perfusion, the ischemic damage marker, lactate, was significantly increased, while glucose levels were significantly decreased. This suggested the conversion of glucose, provided by the Belzer solution, to lactate as in anaerobic glycolysis. In contrast, the glucose levels increased in HOPE+EV during normothermic reperfusion, suggesting that the kidneys have the ability to synthesize glucose through gluconeogenesis, and this can occur since the tissue is well preserved and vital.

On the other hand, the main substrate for renal gluconeogenesis is lactate. Lactate uptake from the tubular lumen occurs through a carrier involving sodium-coupled monocarboxylate transporters (MCTs) expressed by the basolateral side of the tubular proximal cells. Previously, we demonstrated that MCT genes were up-regulated by EV in per-fused rat ischemic kidney [89]. Therefore, our results add another so far unknown and important piece to understanding the effects of EV on ischemic tissue.

In addition, our data show that IRI damage decreased after normothermic reperfusion with HBOCs, which is in contrast with the well-known adverse effects described for HBOC use in vivo (i.e., vasoconstriction, hypertension [90], and oxidative stress [91,92,93,94]). These results have led to disappointing clinical trials [95]. More recently, however, HBOCs have been repurposed as ex vivo perfusion solutions for their capability of providing oxy-gen to isolated organs, a key requirement in NMP technologies [96]. Moreover, although whole blood-based perfusates were initially shown to yield superior preservation of myocardial function in heart ex vivo perfusion in comparison to HBOCs [97], several NMP experiments have shown that HBOCs yield similar, if not better, results in comparison to RBCs in the liver [36,98,99,100] and kidney [37,101,102,103]. In our model, we could not distinguish the contribution from HBOCs and that induced by EV and/or HOPE. Nevertheless, the combined effects are certainly surprising. Some studies affirm that EV effects are temperature-dependent and that low temperatures reduce EV endocytosis by cells [104]. In contrast, in vitro experiments have demonstrated that MSC induce a proliferative effect on tubular epithelial cells even at low temperatures. We confirm here that EV can also act in hypothermia conditions [38].

## 5. Conclusions

In conclusion, in this study we explored an innovative tool for organ preservation aimed at preventing IRI damage in vulnerable marginal kidneys. We demonstrate that EV delivery during HOPE significantly reduces IRI damage in ECD kidneys. Therefore, it can be considered a new organ preservation strategy for increasing the donor pool and improving transplant outcomes. To the best of our knowledge, this is the first report on conditioning human kidneys with MSC-derived EV. Even though the ex vivo model study is limited, as the treated kidneys were not transplanted, we strongly believe that our results may form the basis for future randomized clinical trials.

## Figures and Tables

**Figure 1 biology-11-00350-f001:**
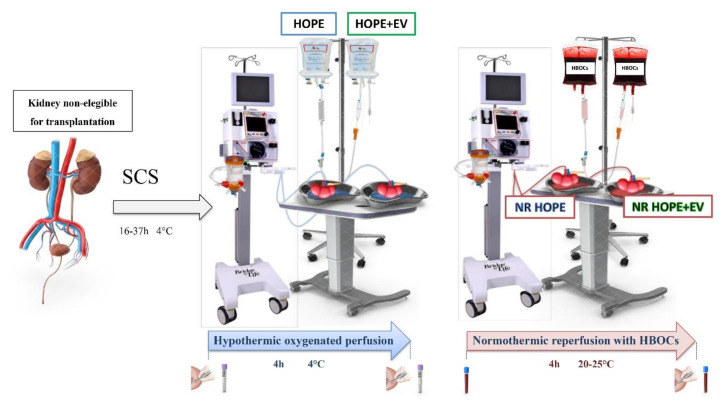
Trial design. Samples of effluent were collected at the beginning and end of HOPE and normothermic reperfusion. Biopsies were performed at the end of hypothermic and normothermic preservations. SCS, static cold storage; HOPE, hypothermic oxygenated machine perfusion; HOPE+EV, hypothermic oxygenated machine perfusion with extracellular vesicles derived from mesenchymal stromal cells; HBOCs, hemoglobin-based oxygen carriers; NR HOPE, normothermic reperfusion of HOPE-perfused kidneys; NR HOPE+EV, normothermic reperfusion of HOPE+EV-perfused kidneys.

**Figure 2 biology-11-00350-f002:**
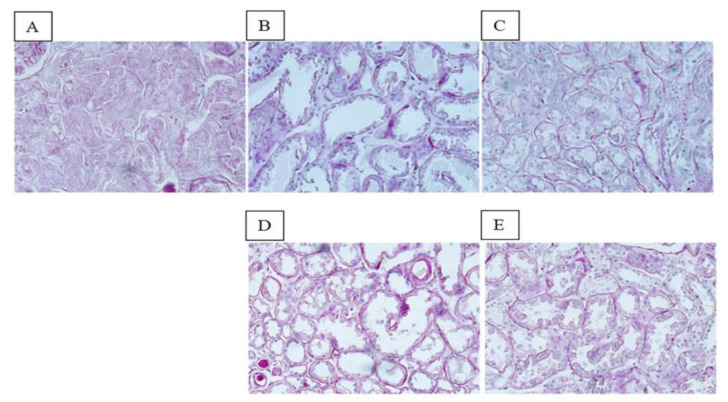
Renal morphology. PAS staining of representative renal sections from the SCS group (**A**), HOPE group (**B**), HOPE+EV group (**C**), NR HOPE group (**D**), and NR HOPE+EV group (**E**). ×20 magnification.

**Figure 3 biology-11-00350-f003:**
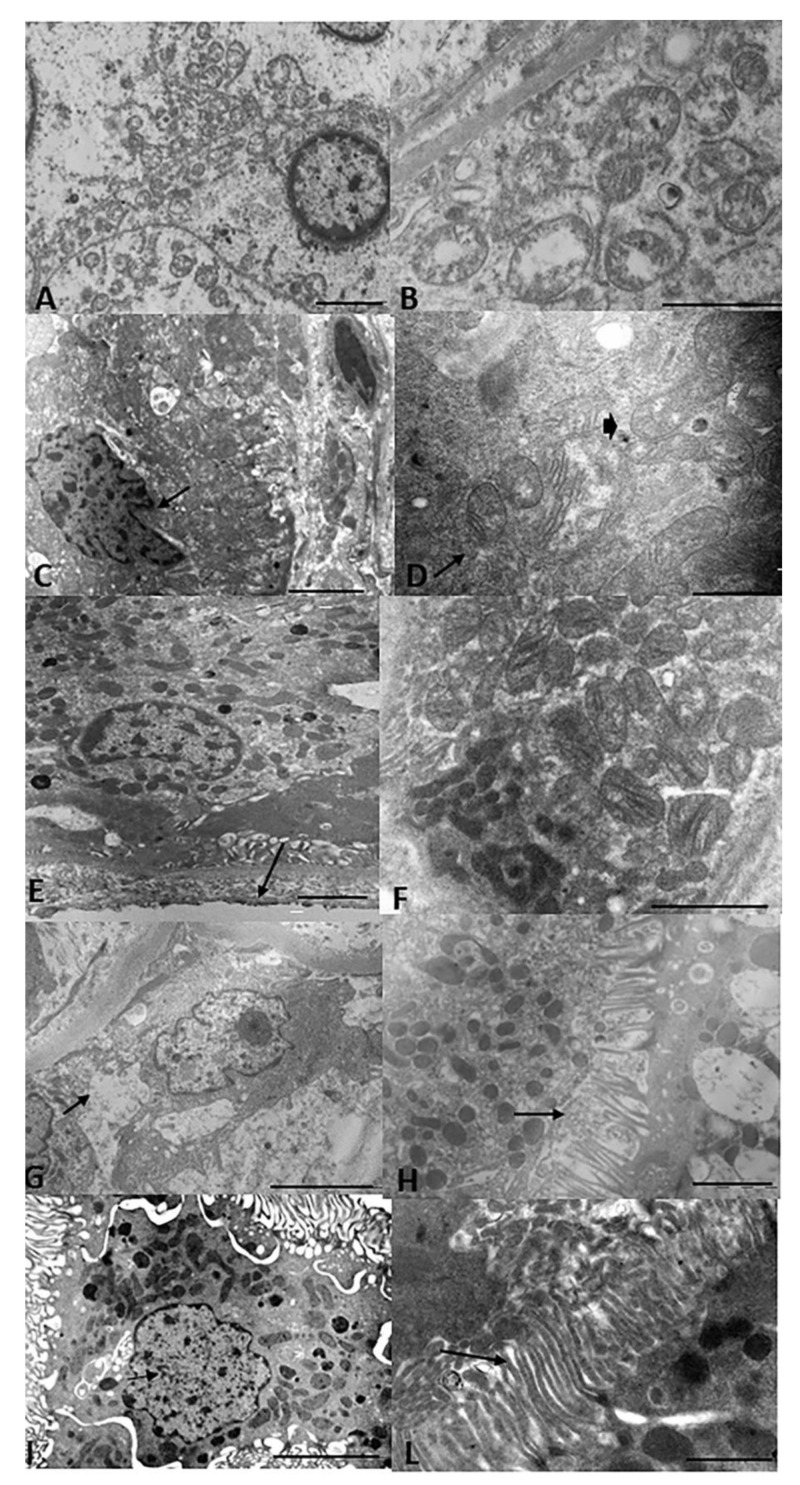
Renal ultrastructure. Representative TEM renal sections of all groups. (**A**) SCS kidney sections showing loss of normal cell architecture, cell shrinkage and necrosis, and thickened chromatin on the nuclear membrane. Bar = 2 µm. (**B**) At greater magnification, dysmorphic and swollen mitochondria with loss or distortion of cristae can be seen. Bar = 2 µm. (**C**) HOPE kidney sections showing partial loss of cell architecture and nuclear membrane invagination. Bar = 2 µm. (**D**) At greater magnification: spheroidal (arrow) and elongated (arrowhead) mitochondria can be seen. Bar = 5 µm. (**E**) HOPE+EV kidney sections showing well-preserved cell architecture, normal nucleus and basal membrane, and numerous basal micropedici (arrow). Bar = 2 µm. (**F**) Higher magnification showing that normal morphology of the mitochondria and cristae was well preserved. Bar = 2 µm. (**G**) NR HOPE kidney sections showing cytoplasmic vacuolization (arrow). Bar = 5 µm. (**H**) At greater magnification, brush borders with short and spaced microvilli (arrow) can be seen. Bar = 2 µm. (**I**) NR HOPE+EV sections showing cells with normal morphology and organelles. Bar = 5 µm. (**L**) At greater magnification, a long, preserved brush border can be seen. Bar = 2 µm.

**Figure 4 biology-11-00350-f004:**
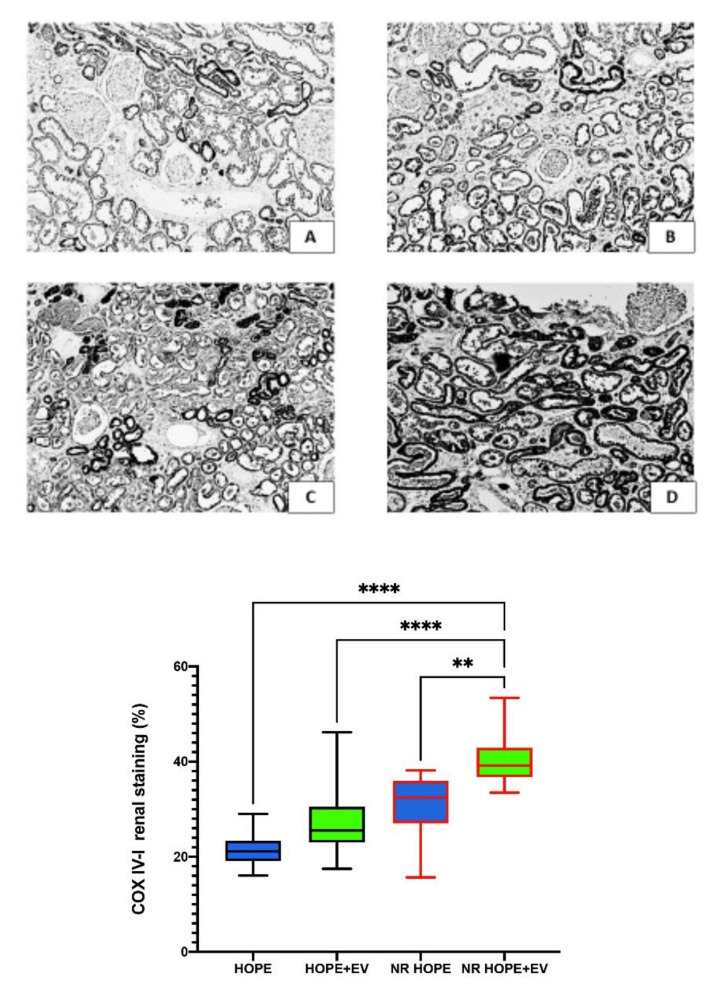
COX IV-1 renal expression. Top: Representative renal sections of COX IV-1 staining in the HOPE group (**A**), HOPE+EV group (**B**), NR-HOPE group (**C**), and NR-HOPE+EV group (**D**). ×20 magnification. Bottom: Columns represent COX IV-1 positive staining percentage in all groups. Data are the median and IQR, or min and max. (NR HOPE+EV vs. HOPE and HOPE+EV **** *p* < 0.0001; vs. NR HOPE ** *p* < 0.005).

**Figure 5 biology-11-00350-f005:**
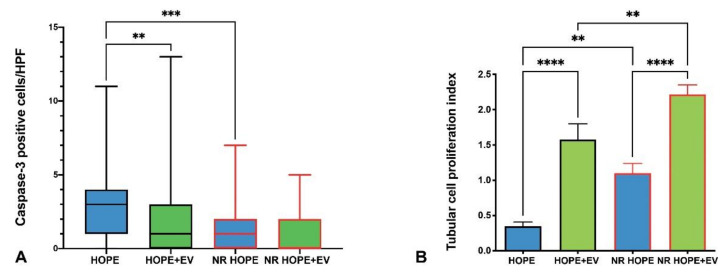
(**A**) Caspase-3 renal expression: columns represent caspase-3-positive cells/HPF in all groups. Data are represented as median and IQR (boxes), and min and max (whiskers). ** *p* < 0.05 vs. HOPE, *** *p* < 0.001 vs. HOPE. (**B**) Tubular cell proliferation index: the number of PCNA-positive nuclei/total nuclei of each tubule was lower in the HOPE and NR HOPE groups than in the HOPE+EV and NR HOPE+EV groups (ANOVA, *p* < 0.0001; Tukey Multiple Comparison Test, HOPE vs. HOPE+EV **** *p* < 0.0001 and vs. NR HOPE ** *p* < 0.005; HOPE+EV vs. NR HOPE+EV *p* < 0.001; NR HOPE vs. NR HOPE+EV *p* < 0.0001).

**Figure 6 biology-11-00350-f006:**
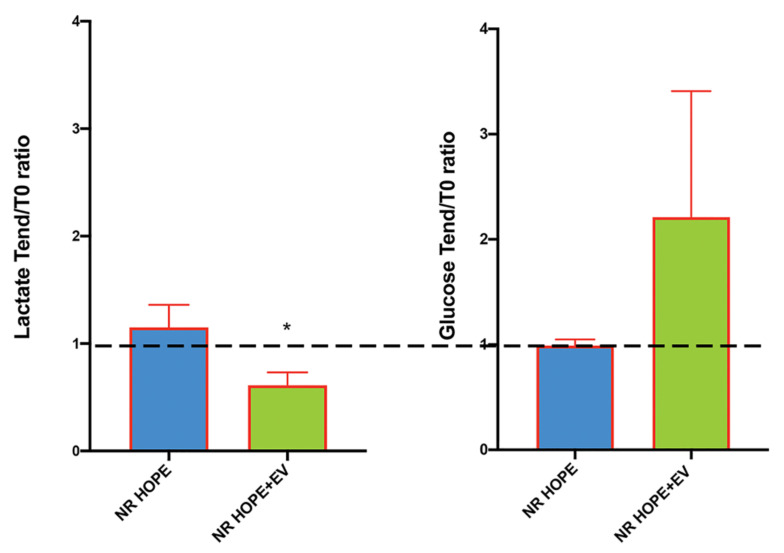
Effluent lactate and glucose levels in all groups at the beginning and end of HOPE and NR HOPE with or without EV, expressed as the Tend/T0 ratio. * *p* < 0.05 vs. NR HOPE.

**Table 1 biology-11-00350-t001:** Donors and kidney characteristics.

	Donor 1	Donor 2	Donor 3	Donor 4	Donor 5
* **Age** *	72	72	80	81	81
* **Sex** *	F	F	M	F	F
***Weight*** (kg)	90	72	85	80	80
***Height*** (cm)	1.55	1.60	1.75	1.50	1.50
***BMI*** (kg/cm^2^)	37.50	28.10	27.80	35.60	35.60
* **Blood group** *	A	0	A	A	0
* **RH BG** *	Pos	Pos	Pos	Pos	Neg
* **Cause of death** *	ch	ch	ch	is	ch
* **Score R kidney** *	6	8	7	6	8
* **Score L kidney** *	6	7	8	7	7
* **Donor type** *	DBD	DBD	DBD	DBD	DBD
***SCS*** (h)	17	37	32	22	16

BMI, body mass index; BG, blood group; R, right; L, left; ch, cerebral hemorrhage; is, ischemic stroke; DBD, donor after brain death; SCS, static cold storage; h, hours.

**Table 2 biology-11-00350-t002:** Perfusion parameters.

Time	Variables Media +/− SD	HOPE	HOPE+EV	NR HOPE	NR HOPE+EV	*p* Value
**T0**	Flow (ml/min)	67.0 +/− 51.2	64.4 +/− 41.9	141.8 +/− 61.3	163.0 +/− 36.3	NS
	Resistances (mmHg min/ml)	0.36 +/− 0.26	0.39 +/− 0.12	0.29 +/− 0.18	0.29 +/− 0.04	NS
**Tend**	Flow (ml/min)	68.3 +/− 30.4	62.2 +/− 24.7	183.0 +/− 71.3	183.3 +/− 36.9	NS
	Resistances (mmHg min/ml)	0.31 +/− 0.13	0.38 +/− 0.12	0.25 +/− 0.13	0.23 +/− 0.04	NS

T0, time at the beginning of HOPE or NR HOPE with and without EV. Tend, time at the end of HOPE or NR HOPE with and without EV.

**Table 3 biology-11-00350-t003:** Lactate and glucose concentrations in effluent fluid at the beginning and end of HOPE with and without EV.

Variables (Media; SD)	HOPE		HOPE+EV		*p* Value
	T0	Tend	T0	Tend	
Lactate (mmol/dL)	0.7 (0.2)	1.4 * (0.5)	0.6 (0.2)	1.9 ° (0.5)	<0.05 * vs T0 HOPE ° vs T0 HOPE+EV
Glucose (mg/dL)	174.3 (4.5)	165.8 (6.9)	184.4 (9.6)	170.2 ^§^ (4.9)	<0.05 ^§^ vs T0 HOPE+EV

* *p* < 0.05 vs. T0 HOPE, ° *p* < 0.05 vs. T0 HOPE+EV; ^§^
*p* < 0.05 vs. T0 HOPE+EV.

## Data Availability

Part of the data presented in this study are available in the Supple- mentary Material. The remaining data presented in this study are available upon reasonable request from the corresponding author.

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
