# Peer review of "Extracellular Vesicles Derived from Mesenchymal Stromal Cells Delivered during Hypothermic Oxygenated Machine Perfusion Repair Ischemic/Reperfusion Damage of Kidneys from Extended Criteria Donors"

_biology, 2022, doi:10.3390/biology11030350_

Round 1
Reviewer 1 Report
What is the purpose of providing a simple summary alongside with an abstract? An abstract by itself should be a simple summary of the study. The provided simple summary is better placed as part of the introduction where a reader usually looks for statements on the novelty of the study, or in the discussion/conclusion to discuss impacts of the results.
Please check misplaced dashes eg. Is-chemia, dys-function, in-crease, extracellu-lar, ..
A graphical abstract should only contain a graphical outline of the study, not titles or complete paragraphs. This can be done better, not only copy-pasting a powerpoint slide. Please extend the graphical portion to full width, remove title and the conlusion statement on the right, so that the labeling in the figure becomes readable.
Line 76: „Mesenchymal Stromal Cells“ → lower case
Line 188+: How much volume of conditioned cell supernatant was used?
Line 191: „detected“ instead „expressed“
Line 276: Why was a COX IV-1 anti-rabbit antibody used against human COX IV-1?
Section Results – 3.2 EV Characterization: When reporting an EV characterization it is essential to show the data instead of writing „data not shown“. Why are not CD73 or CD90 shown as typical MSC markers? What was the concentration of EVs as determined via NTA? In this context, how were these data used to determine a dosing scheme for HOPE+EV and NR+EV? How much EVs were used per treatment?
Figure 5 – why are data differently presented? median/IQR vs min/max. Which statistical test was used and how large is n in each group?
Figure 6 – why is the significance level of differences between groups not indicated in the figure? Which post-hoc test was used? How large is n per group?
Line 451+: Please properly report ANOVA results. Degrees of freedom, F-statistic, post-hoc test etc. None of the different conditions were significantly different, as nothing is indicated on the figure?
Figure 5 and 6 could be combined into one figure with 2 panels.
Line 525: To support the claim that HOPE+EV increases gluconeogenesis, additional experiments such as PEPCK expression analysis or determination of gluconeogenic enzyme activity would be essential, as it can not be exluded at this point that the glucose is released from the kidney cells for whatever reason.
Line 558: The heading „Patents“ is superfluous?
Reviewer 2 Report
The study was aimed to evaluate the impact of combined procedures - hypothermic oxygenated perfusion (HOPE) with and without extracellular vesicles (EV), derived from Mesenchymal Stromal Cells, on ischemic/reperfusion injury of the tissue of kidneys procured from five deceased donors at the age > 70=80 (regarded as the „ marginal donors”), in terms of suitability of these organs for clinical transplantion.
The major conclusion was, that the adding of MSC-derived EV during HOPE procedure significantly reduces the degree of ischemia-reperfusion injury damage in the kidneys procured from marginal deceased donors.
The results have not been verified in a clinical practice, as the kidneys have not been transplantated.
Questions and comments below come from a transplant practitioner.
Questions:
Clinical/technical
- Static cold storage time varied from 16 to 32 hours; was this difference in time relevant in terms of any further result?
- the procedure of obtaining EV seems to be very complex and expensive; is it (technicaly) available on-line, in terms of non-scheduled organ procurement from a deceased donor? The long time of static cold storage suggests the this was required for preperation of EV; is that right?
- 3 of 5 donors were extremely obese (BMI > 35); was it of any importance for the results, as in clinical practice this is a risk factor for delayed graft function?
Legal/ethical
It was marked, that „ Informed Consent Statement: Not applicable”
This is understandable, that donors could not have been asked for a consent, however please clarify the legal issues used in Italy in terms of organ procurement from the deceased-donors (is it a „presumed consent”?)
Here, the kidneys have been procured from patients > age of 70- 80, not to be transplanted (as not eligible by a rule), but to be a subject of a pure scientific Research
Editorial comment: height is probably expressed in (m), not (cm), in table 1
Round 2
Reviewer 1 Report
The authors should clearly outline how many kidneys were investigated per treatment group (HOPE, HOPE+EV, NR HOPE, NR HOPE+EV). These are 4 experimental groups, but there are only 5 donated kidneys investigated. A reader would assume, that only 1 or 2 kidneys were investigated per treatment group. If this is the case, this should be explicitly mentioned as limitation of the study.
Why was exactly 28.5x10^9 EVs chosen as treatment dose? What are the EV concentrations used in similar studies? What proportion of cells in the organ may receive at least on EV at this concentration?
Before publication, the authors should provide high resolution versions of the figures as the labeling of the graphs is not readable, eg. for NTA and the multiplex bead assay data in Supplementary figure 1.
Author Response
The authors should clearly outline how many kidneys were investigated per treatment group (HOPE, HOPE+EV, NR HOPE,NR HOPE+EV)These are 4 experimental groups but there are only 5 donated kidneys investigated. A reader would assume, that only 1 or 2 kidneys were investigated per treatment group. If this is the case, this should be explicitly mentioned as limitation of the study.
We thank the reviewer for the comment. We have integrated the text specifying that the kidneys studied were 10, taken from 5 donors. Both kidneys of the same donor were randomized in a 1: 1 ratio, so five kidneys were perfused with and without EV. Subsequently, all kidneys were reperfused with HBOCs ( NR HOPE and NR HOPE +EV). Therefore each group consisted of 5 kidneys.
Why was exactly 28.5x10^9 EVs chosen as treatment dose? What are the EV concentrations used in similar studies? What proportion of cells in the organ may receive at least on EV at this concentration?
28.5x109 EVs were produced by 3.0x106 human MSC after overnight starvation. The dose was chosen on our previously experience (Gregorini et all, J Cell Mol Med Vol21 N.12,2017 pp3381-3393). In our knowledge there are no other studies that report the dose of EV used to ex vivo perfuse human kidney. We cannot know what is the proportion of cells in human kidneys that receive EVs. We can speculate that EVs might preferentially be up taken by cells in contact with blood vessels; but we don’t have a direct evidence.
Before publication, the authors should provide high resolution versions of the figures as the labeling of the graphs is not readable, eg. for NTA and the multiplex bead assay data in Supplementary figure 1.
We are sorry concerning the not appropriate resolution of figures. We have modified the figures in the revisioned text, and we hope that now they are readable.
